# *RrGT2*, A Key Gene Associated with Anthocyanin Biosynthesis in *Rosa rugosa*, Was Identified Via Virus-Induced Gene Silencing and Overexpression

**DOI:** 10.3390/ijms19124057

**Published:** 2018-12-14

**Authors:** Xiaoming Sui, Mingyuan Zhao, Zongda Xu, Lanyong Zhao, Xu Han

**Affiliations:** Flower Research Laboratory, College of Forestry, Shandong Agricultural University, Taian 271018, China; suixiaomingjiayou@163.com (X.S.); zhaomingy9@163.com (M.Z.); xuzoda@163.com (Z.X.)

**Keywords:** *Rosa rugosa*, *RrGT2* gene, Clone, VIGS, Overexpression, Tobacco, Flower color, Anthocyanin

## Abstract

In this study, a gene with a full-length cDNA of 1422 bp encoding 473 amino acids, designated *RrGT2*, was isolated from *R. rugosa* ‘Zizhi’ and then functionally characterized. *RrGT2* transcripts were detected in various tissues and were proved that their expression patterns corresponded with anthocyanins accumulation. Functional verification of *RrGT2* in *R. rugosa* was performed via VIGS. When *RrGT2* was silenced, the *Rosa* plants displayed a pale petal color phenotype. The detection results showed that the expression of *RrGT2* was significantly downregulated, which was consistent with the decrease of all anthocyanins; while the expression of six key upstream structural genes was normal. Additionally, the in vivo function of *RrGT2* was investigated via its overexpression in tobacco. In transgenic tobacco plants expressing *RrGT2*, anthocyanin accumulation was induced in the flowers, indicating that *RrGT2* could encode a functional GT protein for anthocyanin biosynthesis and could function in other species. The application of VIGS in transgenic tobacco resulted in the treated tobacco plants presenting flowers whose phenotypes were lighter in color than those of normal plants. These results also validated and affirmed previous conclusions. Therefore, we speculated that glycosylation of *RrGT2* plays a crucial role in anthocyanin biosynthesis in *R. rugosa*.

## 1. Introduction

*Rosa rugosa* is an important ornamental plant species that belongs to the genus *Rosa* in the family *Rosaceae*. This species is native to China and is widely distributed worldwide. Because of its unique fragrance, color, cold resistance and drought resistance, there is great potential for the development of this species for use in garden applications. Many varieties of roses exist but most of them are traditional colors such as pink and purple. A few varieties are white and some lack yellow, bright red, orange and compound colors and so forth. [1]. Therefore, the development of innovative rose colors has become the main goal of breeders. At present, scientific research on *R. rugosa* in China and abroad has focused mainly on the development and protection of wild *R. rugosa* resources [2], the analysis of the genetic diversity of *R. rugosa* [3], the optimization of *R. rugosa* essential oil extraction [4], the nutritional value of *R. rugosa* [5], the cultivation and propagation technology of *R. rugosa* [6,7] and strategies for attaining high *R. rugosa* yields. Innovating new *R. rugosa* flower colors has occurred mainly by improving cultivation and management techniques or by trying to cross different varieties during *R. rugosa* breeding; relatively less molecular biology technology has been used in the innovation of new flower colors. However, due to the decrease in wild *R. rugosa* resources and the lack of natural variation in recent years, the existing *R. rugosa* varieties can no longer meet the various needs of gardening applications [1]. However, there is still much room for development in the breeding of new *R. rugosa* varieties via molecular biology. Therefore, studying the mechanism of *R. rugosa* flower color formation and enriching flower color during breeding are urgently needed. Analysis of the pigment composition of rose varieties and the study of the expression characteristics of the key genes encoding enzymes that catalyze the synthesis of rose pigments are important prerequisites for the molecular breeding of rose color traits [8]. Anthocyanins determine the color of higher plant organs. Structural genes (*CHS*, *CHI*, *F3H*, *F3’H*, *DFR*, *ANS*, *3GT*, etc.) and regulatory genes (*MYB*, mostly those of the *R2R3-MYB*, *BHLH* and *WD40* classes) [9,10] related to the anthocyanin biosynthesis pathway have been cloned and sequenced and related protein functional studies have been performed in many plant species, such as petunia, maize, snapdragon and so on [11,12]. However, less anthocyanin-related research has been conducted in rose than in those species.

Flower coloration is caused by the accumulation of pigments comprising mainly flavonoids, carotenoids and betalains. Among them, flavonoids, which comprise chalcones, flavones, flavonols, flavandiols, anthocyanins and proanthocyanidins, are the most important pigments [13,14]. Anthocyanins, which are derived from the anthocyanin biosynthesis pathway, are the largest group of water-soluble plant flavonoids found in the organs of plants, including crop species [15,16,17,18,19]. Anthocyanins are unstable in plants; they exist mainly in the form of glycosides within the vacuole [20]. The flavonoid 3-*O*-glycosyltransferase (*3GT*) gene lies downstream in the anthocyanin synthesis pathway. The enzyme encoded by this gene can catalyze the glycosylation of UDP-glucose to replace the 3 hydroxyl groups of anthocyanin molecules and cause anthocyanin glycosylation to produce colored and stable anthocyanins. Glycosylation can alter the hydrophilicity, biochemical activity and subcellular localization of anthocyanins, which is beneficial to their transport and storage in cells and organisms [21,22,23].

For a long time, GT genes had failed to be cloned and their functions in plant secondary metabolism were unclear. However, since the first cDNA sequence of a GTase was isolated by transposon tagging in maize, an increasing number of GT genes have been cloned and identified [24]. Studies have shown that the *3GT* gene is expressed only in red grape (*Vitis vinifera*) varieties and not in white grape varieties. When a *3GT* transgene was transformed into a colorless embryo, a pale-red bud was produced [25,26]. Studies by Afifi et al. on the expression of five key genes involved in anthocyanin synthesis in grape cell tissue indicate that the presence of the cytotoxic factor eutypine inhibits the expression of *3GT* and thus reduces anthocyanin contents [27]. This finding indicates that *3GT* is the key gene involved in grape skin color (from white to red) and is also a key gene in the anthocyanin biosynthesis pathway [28]. In *Gentiana triflora*, *3GT* expression occurs mostly in blue petals and rarely in white flowers [29]. Expression of the *3GT* gene is very important for anthocyanin accumulation in many plant species and its expression intensity is positively correlated with anthocyanin synthesis [30].

In this study, we cloned and identified the *RrGT2* gene from the petals of *R. rugosa* for the first time. We carried out detailed bioinformatic and homology analyses of the *RrGT2* gene. VIGS results in perennial *Rosa* plants under field conditions suggested that *RrGT2* is related to the biosynthesis of anthocyanins in *R. rugosa*. Stable transformation of the *RrGT2* gene in tobacco showed that its overexpression was positively correlated with the accumulation of anthocyanins. The results of VIGS in transgenic tobacco also confirmed this conclusion. We verified the functions of the *RrGT2* gene in anthocyanin metabolism in both the positive and negative directions to provide useful information for subsequent color-improvement projects in *R. rugosa*.

## 2. Results

### 2.1. Cloning of RrGT2 and Sequence Analysis

In the early stages, we screened the *RrGT2* gene by comparing the differentially expressed genes in the anthocyanin pathway within the *R. rugosa* transcriptome data of our laboratory. The full-length CDS of *RrGT2* (MK034141) was cloned and then confirmed by sequencing [31]. The complete open reading frame from the ATG start codon to the TAA termination codon encodes a 473 amino acid protein (Figure 1A). Multiple sequence alignment (Figure 2) revealed that the *RrGT2* protein, which belongs to the GTB superfamily, displays strong species specificity in the N-terminal region and PSPG conserved domains that consist of 44 amino acid residues in the C-terminal region. A phylogenetic tree (Figure 1B) was constructed from the amino acid sequences of 21 plants, including the sequence of *RrGT2*, using MEGA 5.0 software. The results showed that the *RrGT2* gene shared the highest homology percentages with *FaUGT* (*Fragaria × ananassa*) and *FvGT* (*Fragaria vesca* subsp.), both of which were 89%.

### 2.2. Temporal and Spatial Expression Patterns of RrGT2 in Rosa

Before analyzing the expression patterns of *RrGT2*, we cloned a gene from the cDNA of *R. davurica* with the full-length primers of *RrGT2* and sequenced the gene sequence which was identical to that of *RrGT2*. So, we named it *RdGT2*.

The expression levels of the *RrGT2* and *RdGT2* gene, which significantly differed, were assessed during five flowering stages. In *R. rugosa* ‘Zizhi’ (Figure 3A), the highest expression level of *RrGT2* was observed during the full opening stage and the lowest was observed during the budding stage. In *Rosa davurica* (Figure 3B), the expression level of *RdGT2* was also highest during the full opening stage but lowest during the half opening stage. The expression patterns of the *RrGT2* gene in *R. rugosa* ‘Zizhi’ and *R. davurica* exhibited approximately the same trend.

The expression levels of the *RrGT2* and *RdGT2* gene, which also significantly differed, were assessed in seven different tissue types. The expression level in the leaves, stems and sepals was relatively high but was relatively low in the other tissues in both *R. rugosa* ‘Zizhi’ (Figure 3C) and *R. davurica* (Figure 3D).

### 2.3. VIGS of RrGT2 Reduced the Transcript Abundance of the Endogenous RrGT2 Gene

At 14 days after infection, GFP detection was performed on the newly grown leaves of the infected plants (TRV-GFP and TRV-GFP-*RrGT2*) and on the untreated leaves of both *Rosa* species. GFP imaging (Figure 4A and Figure 5A) showed that the leaves treated with VIGS (TRV-GFP and TRV-GFP-*RrGT2*) showed green fluorescence under longwave ultraviolet light, while the untreated leaves in the control group showed red fluorescence. The corresponding leaves were collected for qRT-PCR detection and the *RrGAPDH* gene was used as an internal control [32] to confirm the efficiency of VIGS. The results (Figure 4D and Figure 5D) showed that the abundance of the *RrGT2* transcript significantly decreased only in the leaves treated with TRV-GFP-*RrGT2* but was expressed normally in the leaves of the plants in the control group and TRV-GFP group.

At 30–35 days after infection, the flowers of *R. rugosa* ‘Zizhi’ changed from the budding stage to the initial opening stage, by which time notable differences in flower color could be observed with the naked eye. The petals in the control group and TRV-GFP group showed no definitive changes in color but the petals in the TRV-GFP-*RrGT2* group were clearly lighter in color (Figure 4B).

At 35–40 days after infection, GFP imaging and qRT-PCR were performed on the blossoming petals of untreated plants and infected plants (TRV-GFP and TRV-GFP-*RrGT2*) of both *Rosa* species. The results were as expected: the petals treated with VIGS (TRV-GFP and TRV-GFP-*RrGT2*) showed green fluorescence under longwave ultraviolet light, while the petals in the control group showed red fluorescence (Figure 4C and Figure 5B). The results of qRT-PCR (Figure 4E and Figure 5E) were consistent with those of the above detection. In both *Rosa* species, the relative expression trends of the *RrGT2* gene were also essentially consistent: the relative expression of the *RrGT2* gene in the TRV-GFP group was essentially the same as that in the control group. However, the transcript abundance of the endogenous *RrGT2* gene in the petals treated with TRV-GFP-*RrGT2* was significantly downregulated.

### 2.4. Reduced Anthocyanin Accumulation in Rosa Petals Was Related to the VIGS of RrGT2

To clarify the role of the posttranscriptional silencing of the *RrGT2* gene in the outcome of the current color change, we compared the relative expression of the six key structural genes, *RrCHS* (KT809351), *RrCHI* (KT809352), *RrF3H* (KT809354), *RrF3’H* (MG735186), *RrDFR* (KT809350) and *RrANS* (KT809353), upstream of the *RrGT2* gene in the anthocyanin pathway (Figure 6). The results showed that the relative expression of the six genes in the control group, TRV-GFP group and TRV-GFP-*RrGT2* group of *R. rugosa* ‘Zizhi’ (Figure 7A) and *R. davurica* (Figure 7B) did not clearly change. Therefore, the effects of these upstream structural genes can be excluded. It is speculated that the change in flower color after VIGS treatment was related to the posttranscriptional silencing of the *RrGT2* gene.

### 2.5. HPLC Analysis of Rosa

The anthocyanin HPLC-generated chromatograms for *R. rugosa* ‘Zizhi’ (Figure 4F) and *R. davurica* (Figure 5F) showed that the components were well separated. Comparisons with standards allowed the contents of different substances to be calculated by their peak area (Appendix A). In ‘Zizhi,’ six kinds of anthocyanins were detected: Cy3G5G, Pg3G5G, Cy3G, Pn3G5G, Pg3G and Pn3G. Pn3G5G had the highest content, while Cy3G5G had the second highest; the contents of the other four anthocyanins were relatively low. Compared with those in the control group and TRV-GFP group, the contents of several anthocyanins in response to the VIGS treatment were clearly reduced. Pn3G5G exhibited the greatest decrease in content, followed by Cy3G5G; the content of Pg3G was no longer detectable (Figure 8A). In *R. davurica*, the six anthocyanins listed above were also detected. However, Cy3G5G had the highest content and Cy3G had the second highest content; the contents of the other four anthocyanins were relatively low. Compared with those in the control group and TRV-GFP group, the contents of the six anthocyanins in response to the VIGS treatment were clearly reduced. Cy3G5G exhibited the greatest decrease in content, followed by Cy3G; no detection of Pn3G was observed (Figure 8B).

### 2.6. Overexpression of RrGT2 Increased the Anthocyanin Accumulation in Tobacco

The *RrGT2* gene was ectopically expressed in tobacco using the binary vector pCAMBIA1304-*RrGT2*. Six independent transgenic tobacco lines that overexpressed the *RrGT2* gene were obtained from Hyg-resistance selection and were cultured under the same conditions. PCR analysis confirmed the presence of the transformed *RrGT2* gene in all the transgenic lines as well as the absence of endogenous *RrGT2* in the tobacco plants of the control group (wild type) and empty vector group (the plants were transformed with an empty pCAMBIA1304 vector) (Figure 9A). qRT-PCR analysis revealed that the expression level of *RrGT2* was significantly higher in the transgenic plants, especially T1, T3 and T6, than in the plants of the control group and empty vector group (Figure 9B). Therefore, those three lines were used for further experiments.

Interestingly, there was no substantial difference in morphology between the transgenic plants and wild-type plants. However, the flower color of the transgenic tobacco lines harboring *RrGT2* was affected; compared with that of the tobacco plants in the control group and empty vector group, the petal pigmentation of the transgenic tobacco plants harboring *RrGT2* was markedly deeper (Figure 9C). This change in corolla color of the transgenic tobacco was already visible prior to anthesis. To confirm that the deeper flower color was attributed to increased pigment levels synthesized from the anthocyanin pathway, the total anthocyanins were determined qualitatively and quantitatively via HPLC. Previous studies have reported that cyanidin-3-*O*-rutinoside mainly exists in the petals of wild-type tobacco [33,34,35]. Even in our study, Cy3G was detected mainly in the flowers of OE-*RrGT2*. Compared with those of the plants in the control group and empty vector group, the contents of anthocyanins in the flowers of T1, T3 and T6 were significantly greater (Figure 8C and Figure 9D and Appendix A). The anthocyanin contents in the three transgenic lines were basically consistent with the trends of the relative expression of *RrGT2*. The expression of *RrGT2* in the transgenic tobacco was clearly correlated with the increased pigmentation observed in the petals.

### 2.7. Expression Patterns of RrGT2 in Transgenic Tobacco

The expression patterns of the *RrGT2* gene in three transgenic tobacco lines were analyzed by qRT-PCR (Figure 9E). The expression levels of *RrGT2*, which significantly differed, were assessed in four different tissue types. The expression levels in the leaves and flowers were relatively high but were relatively low in the stems and roots. Of all the tissue types of the three transgenic lines, the highest gene expression occurred in T3, followed by T1 and the lowest was in T6.

### 2.8. VIGS of RrGT2 Reduced the Anthocyanin Accumulation in Transgenic Tobacco

Previous tests showed that the *RrGT2* transgenic tobacco line T3 had the highest level of gene expression, so it was used as the experimental object of VIGS. At 14 days after infection, GFP detection was performed on the newly grown leaves of tobacco plants in the control group, TRV-GFP group and TRV-GFP-*RrGT2* group. GFP imaging (Figure 10A) showed that the leaves treated with VIGS (TRV-GFP and TRV-GFP-*RrGT2*) showed green fluorescence under longwave ultraviolet light, while the untreated leaves in the control group showed red fluorescence. The corresponding leaves were collected for qRT-PCR detection and the *NtACT* gene was used as an internal control. The results (Figure 10C) showed that the abundance of the *RrGT2* transcript significantly decreased only in the leaves treated with TRV-GFP-*RrGT2* but was expressed normally in the control group and TRV-GFP group. In addition, interestingly, compared with those of the control group and TRV-GFP group, the phenotypes of the leaves of the tobacco plants in the group treated with TRV-GFP-*RrGT2* were similar to those in response to photobleaching.

At 60–75 days after infection, notable differences in flower color could be observed with the naked eye. The flowers in the control group and TRV-GFP group showed no definitive changes in color but the flowers in the TRV-GFP-*RrGT2* group were clearly lighter in color (Figure 10B). The corresponding flowers were collected for qRT-PCR detection and the results (Figure 10D) showed that the abundance of the *RrGT2* transcript significantly decreased only in the flowers treated with TRV-GFP-*RrGT2* but was expressed normally in the control group and TRV-GFP group. To confirm that the lighter flower color was attributed to decreased levels of pigments synthesized from the anthocyanin pathway, the total anthocyanins were determined qualitatively and quantitatively via HPLC. Compared with those in the flowers of the control group and TRV-GFP group, the anthocyanin contents in the VIGS-*RrGT2*-1, -2 and -3 flowers were significantly lower (Figure 8D and Figure 10E and Appendix A), which was essentially consistent with the qRT-PCR results.

## 3. Discussion

At present, research on the flower color of *R. rugosa* requires very innovative and practical studies. Although many genes have been reported to regulate the formation of flower color, few reports on downstream structural genes such as GTs exist. The final formation of anthocyanins depends on the glycosylation of GTs, so it is very important to determine the function and influence of the *RrGT2* gene in *Rosa* color formation. In this study, we successfully cloned the *RrGT2* gene, which had a full-length cDNA of 1422 bp and encoded 473 amino acids, from the petals of *R. rugosa* ‘Zizhi.’

The amino acid sequence alignment between *RrGT2* and GTs from 21 other species indicated that *RrGT2* has a common PSPG motif of the GT superfamily (Figure 2). Previous studies have shown that the conserved PSPG region is related to the substrate recognition and catalytic activity of protein-based enzymes [36,37,38,39,40,41,42]. If the 44 amino acids of the PSPG domain were numbered, those at positions 22, 23 and 44 would play an important role in the selection of enzyme proteoglycan donors. The twenty-second position of tryptophan (Trp, W) can correctly bind UDP-glucose, while arginine (Arg, R) can make UDP-glucuronic acid bind correctly; the twenty-third position of serine (Ser, S) is highly conserved among UDP-glucuronosyltransferases [43,44,45] and the forty-fourth position of glutamine (Gln, Q) and histidine (His, H) is strongly conserved among glucosyltransferases and galactotransferases, respectively [38]. Within the PSPG domain of the *RrGT2* gene, the amino acids at positions 22, 23 and 44 are tryptophan (Trp, W), asparagine (Asn, N) and glutamine (Gln, Q), respectively. Therefore, we speculate that the *RrGT2* gene uses UDP-glucose as the main glycosyl donor and has no glucuronyltransferase activity [45].

The expression of the *RrGT2* gene during floral development and in different tissues was investigated. The expression trends of the *RrGT2* gene differed during different flowering periods, indicating that the expression of the *RrGT2* gene was developmentally regulated during the anthocyanin biosynthesis process. Studies have shown that the accumulation of anthocyanins in red-skinned sand pear, strawberries and litchi is positively correlated with the activity of *UF3GT*. Boss et al. [28] also detected the expression of *UF3GT* in the peels of red grape that accumulated anthocyanin but not in other tissues of red grape or white grape without anthocyanin accumulation. The tissue-specific anthocyanin expression was similar to that of *F3GT* genes in peach, in which expression levels were greatest in tissues with pigment accumulation but relatively low in unpigmented organs [46]. Notably, the stems of both *Rosa* species were purple, which is consistent with the relatively high expression level of the *RrGT2* gene in that tissue. Interestingly, *R. davurica* is one of the parents of *R. rugosa* ‘Zizhi,’ so we speculate that this reason might explain the similar expression patterns between both *Rosa* species. However, with respect to the relatively low expression levels in flowers, this did not mean that the *RrGT2* gene had no effect on flower color formation. We believe that this was the result of using flowers at the budding stage as one of the tissue types. During the budding stage, the expression of the *RrGT2* gene was lowest but during the other stages, it was very high. In addition, *RrGT2* was highly expressed in the leaves and sepals of both *Rosa* species, so we infer that *RrGT2* is also involved in the glycosylation of secondary metabolites in leaves and sepals and plays an important role.

To clarify the role of *RrGT2* in the formation of *R. rugosa* flower color, the VIGS technique was used to specifically silence the *RrGT2* gene in both *Rosa* species as well as to detect and analyze the phenotypes of the flowers. The VIGS system, which involves TRV1 and TRV2, is a powerful tool for use in the functional characterization of genes in vivo [47]. At present, few reports exist about the use of the VIGS system in plant floral organs and most of the tested species thus far have been members of the Solanaceae family. For example, VIGS technology was used to study the genes controlling floral fragrance in *Petunia hybrida* [48] and the roles of the *SlMADSI*, *NbMADS4-1* and *NbMADS4-2* genes in tobacco flowers were also determined via VIGS [49]. Furthermore, the TRV recombinant virus vector was successfully used to induce the silencing of the *CHS* and *GLO1* genes in *Gerbera jamesonii* [50]. In the present study, we developed a VIGS system for use with perennial *Rosa* plants grown naturally in the field as experimental materials for the first time and we used this system to study key genes of *Rosa* color and obtained a preliminary result of gene silencing efficiency (Appendix A) and other relevant results. Compared with the control conditions, the conditions resulting from the established optimal VIGS system resulted in clearly lighter petal color of both *Rosa* species, which was consistent with the significantly downregulated transcript abundance of the endogenous *RrGT2* gene. The relative expression of the six key upstream structural genes (*RrCHS*, *RrCHI*, *RrF3H*, *RrF3’H*, *RrDFR* and *RrANS*) remained unchanged. In the anthocyanin biosynthesis pathway, the upstream genes are precursors for anthocyanin biosynthesis [29]. Therefore, silencing of the *RrGT2* gene might lead to such a change.

The contents of eight anthocyanins, Cy3G, Cy3G5G, Pg3G, Pg3G5G, Pn3G, Pn3G5G, Dp3G and Dp3G5G, were analyzed qualitatively and quantitatively via HPLC. The results showed that the most abundant anthocyanin in the petals of *R. rugosa* ‘Zizhi’ was Pn3G5G, which is consistent with the results of Zhang et al. [51]. The content of Cy3G5G was the second highest and the other anthocyanin contents were relatively low; no presence of Dp3G or Dp3G5G was detected. With respect to *R. davurica*, this is the first time different kinds and contents of anthocyanins were detected in the petals. The Cy3G5G content was predominant; that is, the coloration of the *R. davurica* petals was affected mainly by Cy3G5G, while the other anthocyanins contributed little to flower color. After performing the VIGS treatment, we again analyzed both *Rosa* species via HPLC. The results showed that the contents of all the different kinds of anthocyanins decreased to some extent and that the decrease in the contents of several major anthocyanins was clear in both *Rosa* species. These results are in agreement with the lighter flower color phenotypes and the relatively downregulated expression level of the endogenous *RrGT2* gene in response to VIGS treatment. Therefore, it can be inferred that *RrGT2* is a key structural gene that directly affects the formation of anthocyanins in *R. rugosa*.

In plant secondary metabolism, numerous glycosides have already been isolated as biologically active compounds and some of them have been widely used as important medicines. GTs usually act in the final stages of plant secondary metabolism and are used for stabilizing and solubilizing various low-molecular-mass compounds, such as flower pigments [52,53] and for regulating the action of functional compounds such as plant hormones [54,55,56]. To date, most studies on the characteristics of GT enzymes have been derived from recombinant proteins produced in bacterial cells and characterized in vitro. However, very few published studies exist on the characterization of GTs in vivo. To investigate the function of the *RrGT2* gene in anthocyanin biosynthesis in vivo, *RrGT2* was first transferred into tobacco, which enhanced the flower coloration of the transgenic tobacco plants. In addition, after analyzing the tissue-specific expression of various transgenic tobacco lines, we found that the *RrGT2* gene was highly expressed not only in the flowers but also in the leaves. This phenomenon was consistent with the results of tissue-specific expression analysis in *Rosa*. Therefore, we speculated that the *RrGT2* gene may play an important role in the regulation of the growth and development of leaves, even whole plants, in addition to the anthocyanin biosynthesis pathway. However, whether in *R. rugosa* or in tobacco, the transcriptional regulatory mechanisms related to *RrGT2* expression patterns are still unclear, which requires further study in the future. Anyway, the results of the transgenic experiments proved that the exogenous *RrGT2* enzymes could also affect the synthesis of anthocyanins in different species. In other words, the function of *RrGT2* in anthocyanin biosynthesis and other aspects can be exchanged among plant species.

In this study, we used VIGS technology twice to explore the function of the *RrGT2* gene. Notably, we have applied VIGS technology to transgenic tobacco. Compared with the results of the transgenic experiments, the results of these VIGS experiments successfully verified the function of *RrGT2* in tobacco in the reverse direction. As far as we know, no such reports currently exist, so this study remains novel. In our experiments, we observed that the tobacco flowers were significantly pale in color when the *RrGT2* gene was successfully silenced. As determined via HPLC detection, this phenomenon was mostly due to decreased levels of Cy3G, which is synthesized from the anthocyanin pathway. In addition, we also observed that the phenotypes of the leaves of tobacco plants whose *RrGT2* gene was silenced were similar to those in response to photobleaching. This phenomenon was consistent with the above results of tissue-specific expression analysis of the *RrGT2* gene in *Rosa* and transgenic tobacco. This undoubtedly confirmed our previous inference about the function of the *RrGT2* gene in leaves.

## 4. Materials and Methods

### 4.1. Plant Materials

With respect to *Rosa*, *R. rugosa* ‘Zizhi’ and *R. davurica* plants cultivated in the *Rosa* germplasm nursery of Shandong Agricultural University were used as test materials. We collected petals at the budding stage, initial opening stage, half opening stage, full opening stage and wilting stage as well as seven different tissue samples (roots, stems, leaves, sepals, stamens, pistils and petals at the budding stage) in the mornings of sunny days from 20 April to 10 May 2017. After they were flash frozen in liquid nitrogen, all samples, which were collected in triplicate, were put into a −80 °C refrigerator for storage.

With respect to tobacco, wild-type plants were used as transgenic materials. After the tobacco seeds were disinfected by soaking in 70–75% ethanol for 2 min, rinsing with aseptic water once, soaking with 3.5% NaClO for 10–15 min and then rinsing with aseptic water 5 times, they were sown in Murashige and Skoog (MS) solid medium (without antibiotics). After 3 days of vernalization at 4 °C in darkness, the seeds were placed in a growth chamber (25 °C, 16 h/23 °C, 8 h day/night, 60% relative humidity) for approximately 30 days. The germless tobacco seedlings that grew well were selected as follow-up experimental materials.

### 4.2. Extraction of Total RNA and Synthesis of First-Strand cDNA

Total RNA was extracted via an EASY Spin Plant RNA Rapid Extraction Kit (Aidlab Biotech, Beijing, China) in accordance with the manufacturer’s specifications. The integrity of the RNA was measured by gel electrophoresis with 1.0% nondenatured agarose, the purity and concentration of the RNA were detected by a Nanodrop 2000C ultra-microspectrophotometer (Thermo Fisher Scientific, Wilmington, DE, USA) and the qualified RNA was preserved at −80 °C. First-strand cDNA was synthesized via a 5× All-In-One RT MasterMix Reverse Transcription Kit (ABM Company, Vancouver, Canada) in accordance with both the manufacturer’s protocol and the requirements of RT-PCR and qRT-PCR.

### 4.3. Cloning of the Full-Length CDS of RrGT2

We identified the *RrGT2* gene that contained the complete 5′ CDS from the *R. rugosa* transcriptome data in our laboratory. The cDNA 3′ terminal sequence of the target gene was then amplified by 3′-RACE technology. *RrGT2*-F and *RrGT2*-R primers (Appendix A) were designed and amplified according to the full-length cDNA sequence of the *RrGT2* gene [31].

### 4.4. Tobacco Stable Transformation

The plasmids of *pCAMBIA1304* vectors and the *RrGT2* gene with restriction sites (*SpeI* and *BstEII*) were extracted and digested by two enzymes. The digestion products were then ligated with DNA ligase (Appendix A) and transformed into *Agrobacterium tumefaciens*.

*A. tumefaciens*-mediated leaf disc transformation [57] was used to transform tobacco. First, the leaves of the cultured tobacco sterile seedlings were pruned to the appropriate size. *A. tumefaciens* infection was carried out after 2 days of preculture. Acetosyringone (AS) was added to the infective liquid and an empty *pCAMBIA1304* vector was used as a control. After infection, the plants were subjected to a dark treatment for 2 days, after which they were cultured in a growth chamber (25 °C, 16 h/23 °C, 8 h day/night, 60% relative humidity). Differentiation culture and rooting culture were carried out on MS media supplemented with relevant hormones and antibiotics. Hygromycin (Hyg) was used for screening resistant seedlings. The plantlets exhibiting good growth potential were transplanted into small flowerpots that contained substrate after seedling refining and the original growth environment was maintained.

### 4.5. VIGS in Rosa

On the basis of a modified TRV-GFP vector, a TRV-GFP-*RrGT2* recombinant viral vector was constructed. pTRV1 and pTRV2-GFP are two RNA strands of the TRV-GFP virus vector and the multiple cloning sites are mainly within pTRV2-GFP. For silencing *RrGT2* specifically in *R. rugosa*, a 406 bp fragment of the *RrGT2* gene was amplified and cloned into pTRV2-GFP (Appendix A).

The pTRV1, pTRV2-GFP and pTRV2-GFP-*RrGT2* plasmids were transformed into *A. tumefaciens*, which was then cultured in YEB media that contained kanamycin, rifampicin and AS at 28 °C for 14–16 h until an OD_600_ = 1.5 was reached. Before infection, pTRV1 was added to the infection liquid that contained pTRV2-GFP and pTRV2-GFP-*RrGT2* in equal volume; the solution was subsequently mixed, forming a complete TRV-GFP and TRV-GFP-*RrGT2* virus carrier. The mixed bacterial solution was kept at room temperature in darkness for 4 h [58,59,60,61,62,63].

Perennial *Rosa* plants that grew naturally in the field were used as experimental materials and the experimental treatment time (from mid-March to mid-April) was approximately one month before *R. rugosa* flowering. And before setting an inoculation date, we will look into the weather for at least a week to avoid bad weather. In addition, we set the specific inoculation time between 14:00 and 16:00 in the afternoon, because during this time, the environment temperature is relatively high, which is more in line with the inoculation operation of VIGS. Another reason is that inoculation in the afternoon leads to a faster transition to night, making dark processing more real. Because it was difficult to inject the leaves and twigs with syringes and because vacuum infiltration could not be used in the field, we used the method that involved first scratching the leaves and twigs and then infecting them with *A. tumefaciens*. To improve the infection efficiency, 0.01% Silwet L-77 was added to the infection liquid and the plants were subjected to darkness for 24 h after infection for 10 min.

### 4.6. VIGS in Transgenic Tobacco

The final concentration of the virus infective fluid needed to reach OD_600_ = 1.0. In addition, the other aspects of the preparation of the virus infection solution and other preliminary preparation works were consistent with the above methods. First, we selected one of several *RrGT2* transgenic tobacco lines as the experimental object of VIGS. Tobacco infiltration was then performed as described by Liu et al. [62]. The *A. tumefaciens* cultures containing pTRV1 and pTRV2 or their derivatives (1:1, *v*/*v*) were injected into the lowest leaf of four-leaf stage plants by using a 1 mL needleless syringe. After infiltration, the tobacco plants were subjected to a dark treatment for 12 h, after which they were cultured in a growth chamber (25 °C, 16 h/23 °C, 8 h day/night, 60% relative humidity).

### 4.7. GFP Imaging

The detection and imaging of the visualized GFP in *Rosa* and transgenic tobacco plants after VIGS treatment were performed at night with a handheld high-intensity ultraviolet lamp (Model SB-100P/F; Spectronics Corporation, Westbury, NY, USA) and a Nikon D90 camera, respectively.

### 4.8. qRT-PCR Detection

We analyzed the gene expression by qRT-PCR on a Bio-Rad CFX96^TM^ Real-Time PCR instrument (Bio-Rad, Inc., Philadelphia, PA, USA). The qRT-PCR mixture (total volume of 20 μL) contained 10 μL of SYBR^®^ Premix Ex Taq™ (TaKaRa, Inc., Kusatsu, Japan), 8.2 μL of ddH_2_O, 0.4 μL of each primer and 1 μL of cDNA. The PCR program consisted of an initial step of 95 °C for 30 s; 40 cycles of 95 °C for 5 s and 60 °C for 30 s; and then a dissociation stage of 95 °C for 10 s, 65 °C for 5 s and 95 °C for 5 s. Each gene was assessed via three biological replicates. The relative expression levels of the genes were calculated by the 2^−ΔΔCt^ method [64].

### 4.9. Total Anthocyanin Extraction and HPLC Analysis

All samples (0.1 g fresh weight) were homogenized in liquid nitrogen, after which they were extracted with 5 mL of an acidic methanol solution (70:0.1:29.9, *v*/*v*/*v*; CH_3_OH:HCl:H_2_O) at 4 °C in darkness for 24 h and then sonicated for 30 min [65]. After centrifugation, each extract was passed through a membrane filter (0.22 mm).

Qualitative and quantitative analyses of anthocyanins were performed via HPLC. The chromatographic analysis was conducted using a Prominence LC-20AT series HPLC system (Shimadzu, Inc., Kyoto, Japan) with a detection wavelength of 530 nm and the column (TC-C18 column, 5 µm, 4.6 mm × 250 mm) was maintained at 30 °C. The eluent consisted of an aqueous solution A (0.1% formic acid in water) and organic solvent B (acetonitrile). The gradient elution program was modified as described previously [66]: 0 min, 10% B; 15 min, 17% B; 20 min, 23% B; 25 min, 23% B; and 30 min, 10% B. Moreover, the eluent flow rate was 1.0 mL/min, with a 10 µL injection volume. Cy3G, Cy3G5G, Pg3G, Pg3G5G, Pn3G, Pn3G5G, Dp3G and Dp3G5G (EXTRASYNTHESE Trading Company, Lille-Lezennes, France) were used as references for anthocyanin analysis. Three independent biological replicates were measured for each sample.

### 4.10. Statistical Analyses

Three independent biological replicates were measured for each sample and the data presented as the mean ± standard error (SE). Where applicable, data were analyzed by Student’s *t* test in a two-tailed analysis. Values of *p* < 0.05 or <0.01 were considered to be statistically significant.

## 5. Conclusions

In conclusion, the *RrGT2* gene from *R. rugosa* was successfully cloned and characterized. Our results demonstrated that *RrGT2* has all the conserved amino acid residues that are typical of the GT enzyme. Transcript analysis revealed that *RrGT2* was expressed in specific tissues and was developmentally regulated, suggesting that *RrGT2* might act as a modified enzyme in the anthocyanin biosynthesis pathway. The functional verification of *RrGT2* in *Rosa* via VIGS revealed that *RrGT2* is a key structural gene that directly affects the formation of anthocyanins in *R. rugosa*. By overexpressing *RrGT2* in tobacco, we found an increase in anthocyanins in flowers, indicating that *RrGT2* encodes a functional GT protein for anthocyanin glucosylation and could function in other species. Furthermore, VIGS of the *RrGT2* gene in transgenic tobacco resulted in a decrease in the total content of anthocyanins that accumulated in the flowers, which further confirmed that *RrGT2* is involved in the modification of flower color.

## Figures and Tables

**Figure 1 ijms-19-04057-f001:**
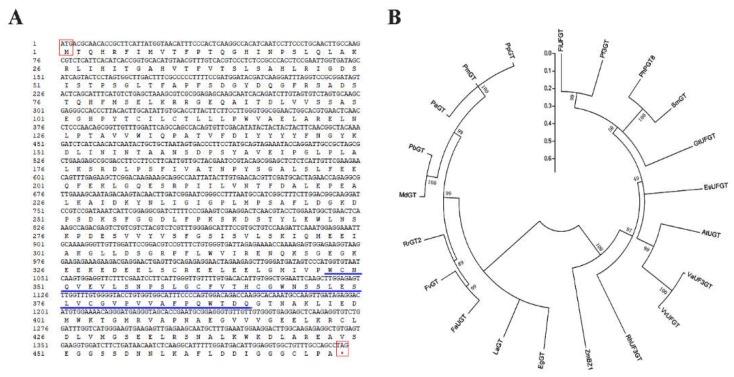
cDNA sequence analysis and phylogenetic tree analysis of the *RrGT2* gene. (**A**) cDNA sequence of *RrGT2* and its deduced amino acids. The red box shows the start codon and the termination codon as well as the amino acids they encode. The PSPG domains are underlined by the blue line. (**B**) Phylogenetic tree of amino acid sequences of *RrGT2* and *GT* members from other plant species. The tree was constructed by the neighbor-joining method using MEGA 5.0 software. The branch numbers represent the percentage of bootstrap values from 1000 sampling replicates and the scale indicates the branch lengths. The gene names from various plant species and the NCBI GenBank accession numbers for the sequences are as follows: *RhUF3GT* (AB599928.1) from a *Rosa* hybrid cultivar; *AtUGT* (UGTNM-121711) from *Arabidopsis thaliana*; *EsUFGT* (KJ648620) from *Epimedium sagittatum*; *FaUGT* (KP337600.1) from *Fragaria* × *ananassa*; *FiUFGT* (AF127218.1) from *Forsythia* × *intermedia*; *FvGT* (XM_004298174.2) from *Fragaria vesca* subspecies; *GtUFGT* (D85186.1) from *Gentiana triflora*; *LaGT* (XM_019560329.1) from *Lupinus angustifolius*; *MdGT* (XM_008350196.2) from *Malus* × *domestica*; *PaGT* (XM_021963118.1) from *Prunus avium*; *PbGT* (XM_009339472.2) from *Pyrus* × *bretschneideri*; *Pf3GT* (AB002818) from *Perilla frutescens*; *PhPGT8* (AB027454) from *Petunia* × *hybrida*; *PmGT* (XM_008224513.2) from *Prunus mume*; *PpGT* (XM_007221257.2) from *Prunus persica*; *SmGT* (X77369.1) from *Solanum melongena*; *VaUF3GT* (FJ169463.1) from *Vitis amurensis*; *VvUFGT* (AF000371) from *Vitis vinifera*; *ZmBZ1* (NM_001112416.1) from *Zea mays*; and *EgGT* (XM_012988948.1) from *Erythranthe guttatus*.

**Figure 2 ijms-19-04057-f002:**
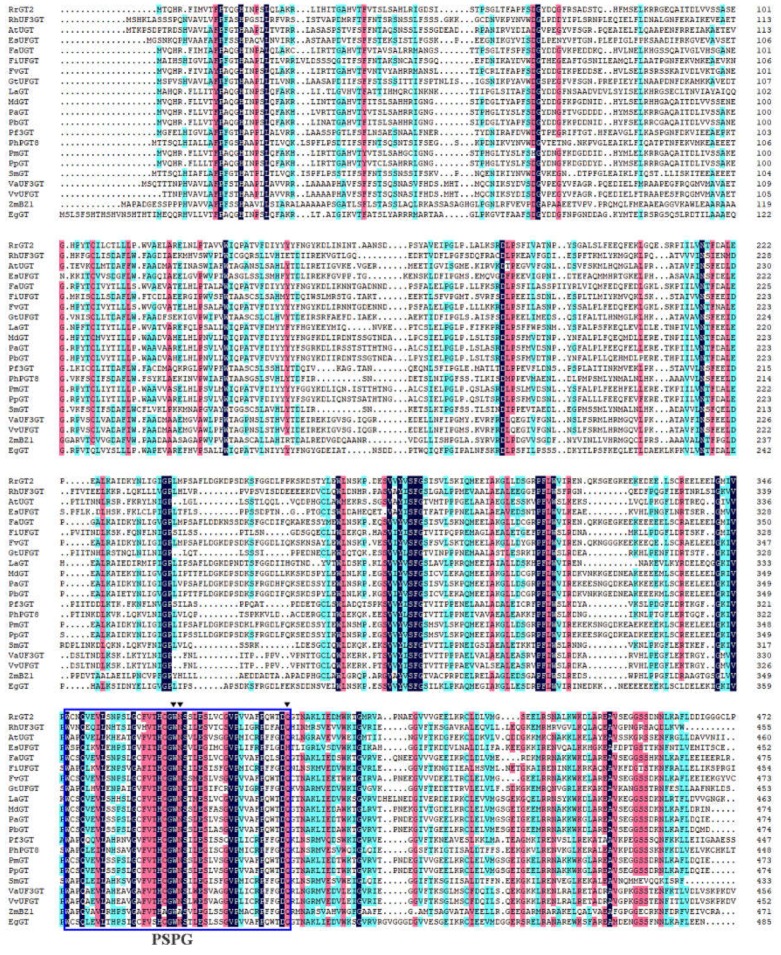
Amino acid sequence homology analysis of the *RrGT2* gene and GTs from other species. Alignments were performed using DNAMAN (version 6.0). The blue box shows the PSPG domains. The black triangles (from left to right) indicate the 22nd, 23rd and 44th amino acids in the PSPG box. The accession numbers are the same as those in Figure 1B.

**Figure 3 ijms-19-04057-f003:**
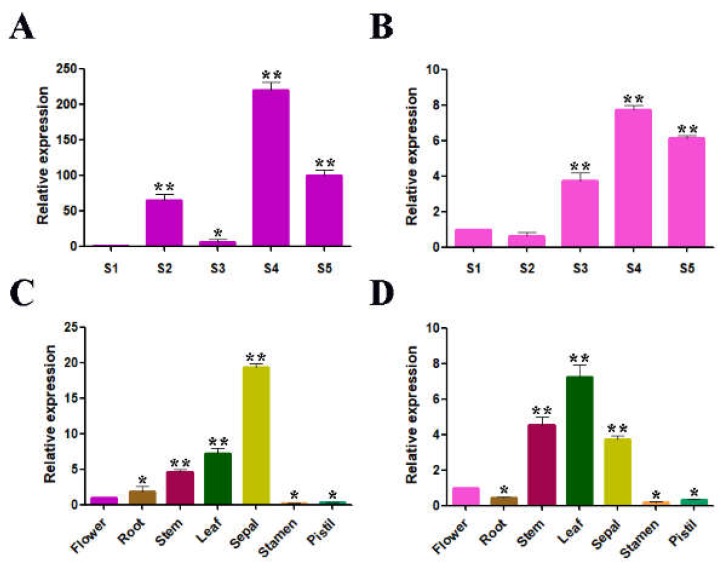
Temporal and spatial expression patterns of *RrGT2*. Relative expression of the *RrGT2* and *RdGT2* gene during the five flowering stages of *R. rugosa* ‘Zizhi’ (**A**) and *R. davurica* (**B**). S1, budding stage; S2, initial opening stage; S3, half opening stage; S4, full opening stage; S5, wilting stage. Relative expression of the *RrGT2* and *RdGT2* gene in seven different tissues of *R. rugosa* ‘Zizhi’ (**C**) and *R. davurica* (**D**). The error bars represent the SDs of triplicate reactions. The experiment was repeated three times and each yielded similar results. * *p* < 0.05 and ** *p* < 0.01 indicate significant differences between different flowering stages and between different tissue types.

**Figure 4 ijms-19-04057-f004:**
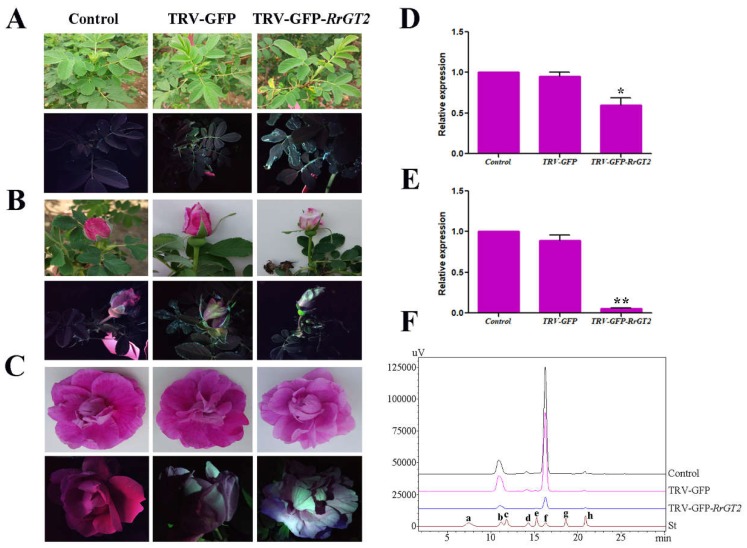
Validation of VIGS in *R. rugosa* ‘Zizhi.’ Comparisons between the control group and VIGS-treated groups (TRV-GFP and TRV-GFP-*RrGT2*) with respect to leaves (**A**), flowers between budding stage and initial opening stage (**B**) and flowers at the full opening stage (**C**). The plants were imaged under normal light and ultraviolet illumination. qRT-PCR detection of leaves (**D**) and flowers at the full opening stage (**E**) in the control and VIGS-treated groups. *RrGAPDH* was used as an internal control. The error bars represent the SDs of triplicate reactions. The experiment was repeated three times and each yielded similar results. * and ** indicate a significant difference from the control at *p* < 0.05 and *p* < 0.01, respectively, according to Student’s *t*-test. (**F**) HPLC-derived chromatograms of flowers at the full opening stage. Eight kinds of anthocyanin standards (St) were used for detection: (a) Dp3G5G; (b) Cy3G5G; (c) Dp3G; (d) Pg3G5G; (e) Cy3G; (f) Pn3G5G; (g) Pg3G; and (h) Pn3G.

**Figure 5 ijms-19-04057-f005:**
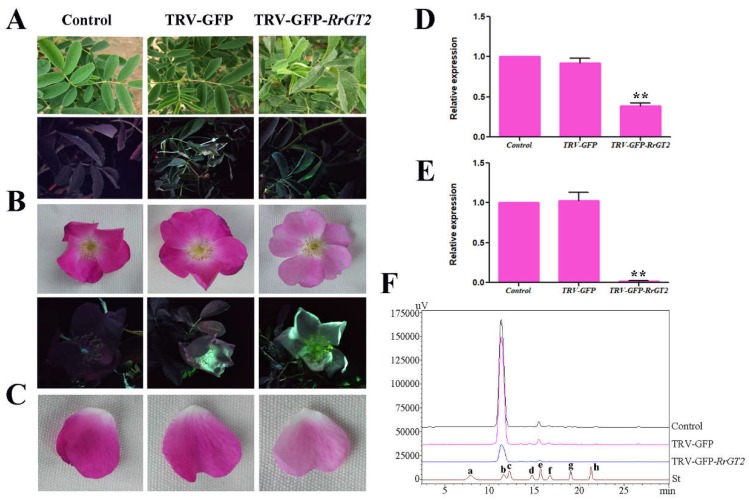
Validation of VIGS in *R. davurica*. Comparisons between the control group and VIGS-treated groups (TRV-GFP and TRV-GFP-*RrGT2*) with respect to leaves (**A**) and flowers at the full opening stage (**B**). The plants were imaged under normal light and ultraviolet illumination. (**C**) The color contrast of a single petal. qRT-PCR detection of leaves (**D**) and flowers at the full opening stage (**E**) in the control and VIGS-treated groups. *RrGAPDH* was used as an internal control. The error bars represent the SDs of triplicate reactions. The experiment was repeated three times and each yielded similar results. ** indicates a significant difference from the control at *p* < 0.01 according to Student’s *t*-test. (**F**) HPLC-derived chromatograms of flowers at the full opening stage. Eight kinds of anthocyanin standards (St) were used for detection: (a) Dp3G5G; (b) Cy3G5G; (c) Dp3G; (d) Pg3G5G; (e) Cy3G; (f) Pn3G5G; (g) Pg3G; and (h) Pn3G.

**Figure 6 ijms-19-04057-f006:**
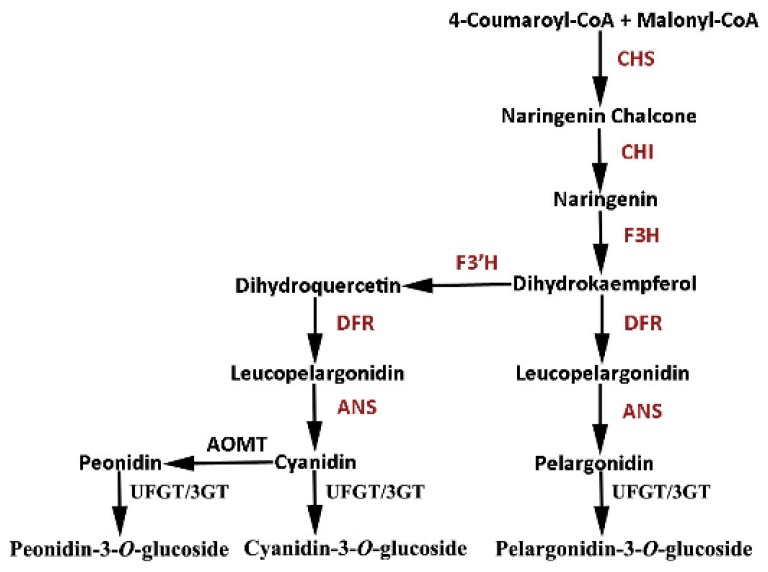
The metabolic pathway analysis of structural genes involved in anthocyanin biosynthesis of *R. rugosa.*

**Figure 7 ijms-19-04057-f007:**
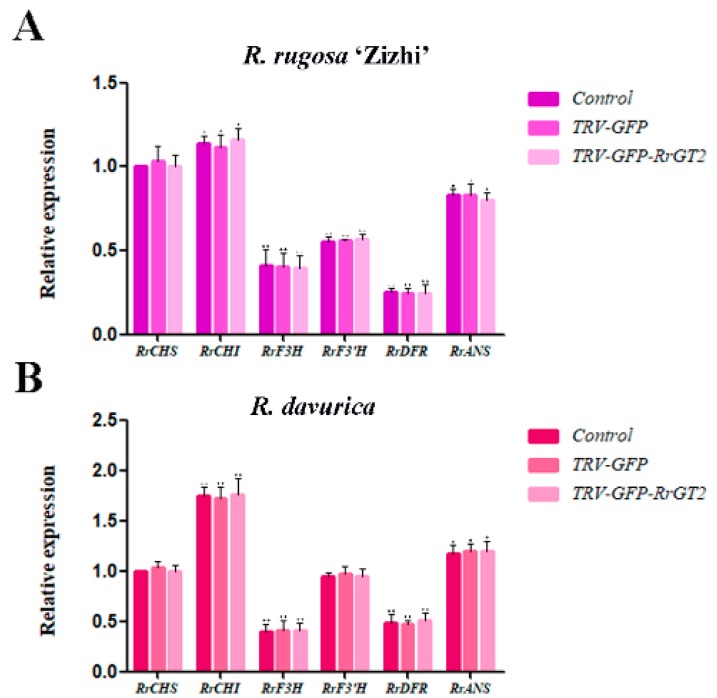
Relative expression levels of six key structural genes (*RrCHS*, *RrCHI*, *RrF3H*, *RrF3’H*, *RrDFR* and *RrANS*) upstream of *RrGT2* in the anthocyanin pathway of *R. rugosa* ‘Zizhi’ (**A**) and *R. davurica* (**B**) at the full opening stage. *RrGAPDH* was used as the internal control. The error bars represent the SDs of triplicate reactions. The experiment was repeated three times and each yielded similar results. * and ** indicate a significant difference from the relative expression levels of *RrCHS* in the control group at *p* < 0.05 and *p* < 0.01, respectively, according to Student’s *t*-test.

**Figure 8 ijms-19-04057-f008:**
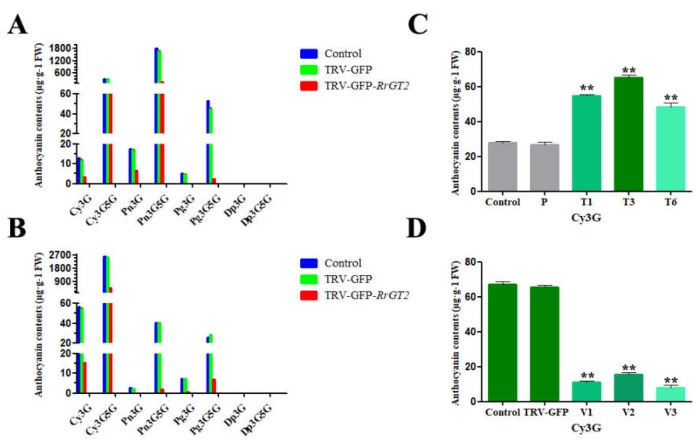
Comparative analysis of anthocyanin components and contents. Anthocyanin components and contents in the flowers of *R. rugosa* ‘Zizhi’ (**A**) and *R. davurica* (**B**) subjected to different VIGS treatments. (**C**) Anthocyanin component and contents in the flowers of transgenic tobacco. (**D**) Anthocyanin component and contents in the flowers of transgenic tobacco subjected to different VIGS treatments. ** indicates a significant difference from the control at *p* < 0.01 according to Student’s *t*-test.

**Figure 9 ijms-19-04057-f009:**
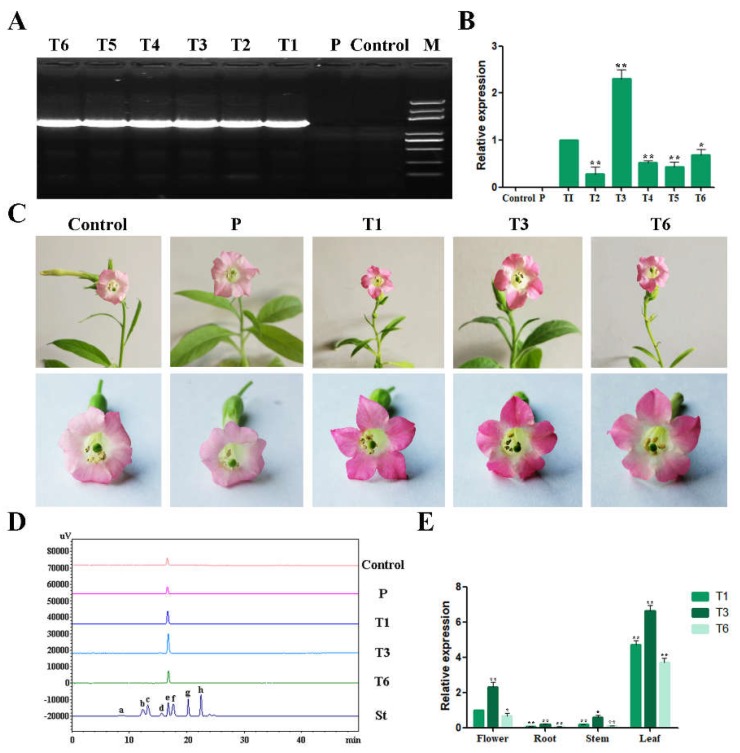
Analysis of tobacco lines overexpressing the *RrGT2* gene. (**A**) The results of positive PCR detection in transgenic tobacco lines. (**B**) The results of qRT-PCR detection in transgenic tobacco lines. (**C**) Phenotypic comparison of flowers between the transgenic tobacco group and the control group and empty vector group. (**D**) HPLC-derived chromatograms of the flowers of the control group, empty vector group and three transgenic tobacco lines. Eight kinds of anthocyanin standards (St) were used for detection: (a) Dp3G5G; (b) Cy3G5G; (c) Dp3G; (d) Pg3G5G; (e) Cy3G; (f) Pn3G5G; (g) Pg3G; and (h) Pn3G. (**E**) Temporal and spatial expression patterns of *RrGT2* in the three transgenic tobacco lines. The error bars represent the SDs of triplicate reactions. The experiment was repeated three times and each yielded similar results. * and ** indicate a significant difference in the relative expression levels at *p* < 0.05 and *p* < 0.01, respectively, according to Student’s *t*-test.

**Figure 10 ijms-19-04057-f010:**
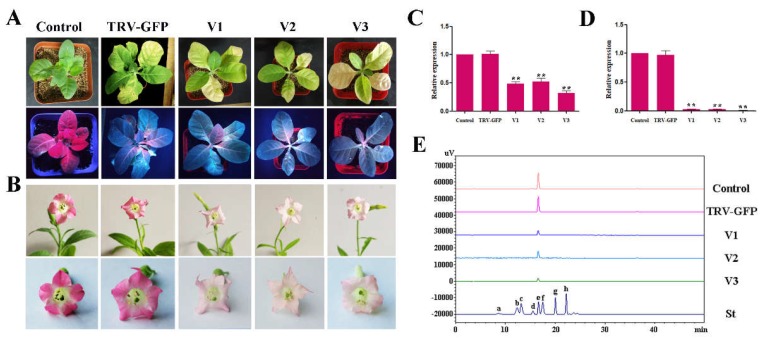
Analysis of OE-*RrGT2* tobacco plants treated with VIGS. (**A**) Comparisons between the control group and the VIGS-treated groups (TRV-GFP and TRV-GFP-*RrGT2*) with respect to leaves at 14 days after infection. The plants were imaged under normal light and ultraviolet illumination. V1, VIGS-*RrGT2*-1; V2, VIGS-*RrGT2*-2; V3, VIGS-*RrGT2*-3. (**B**) Phenotypic comparison of the flowers between the OE-*RrGT2*-VIGS tobacco group and the control group and empty vector group. qRT-PCR detection in the leaves (**C**) and flowers (**D**) of the control and VIGS-treated groups. The error bars represent the SDs of triplicate reactions. The experiment was repeated three times and each yielded similar results. ** indicates a significant difference from the control at *p* < 0.01 according to Student’s *t*-test. (**E**) HPLC-derived chromatograms of the flowers in the control group and VIGS-treated groups. Eight kinds of anthocyanin standards (St) were used for detection: (a) Dp3G5G; (b) Cy3G5G; (c) Dp3G; (d) Pg3G5G; (e) Cy3G; (f) Pn3G5G; (g) Pg3G; and (h) Pn3G.

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
