# Peer review of "RrGT2*, A Key Gene Associated with Anthocyanin Biosynthesis in *Rosa rugosa*, Was Identified Via Virus-Induced Gene Silencing and Overexpression"

_ijms, 2018, doi:10.3390/ijms19124057_

Reviewer 1 Report

Sui et al. present results from virus-induced gene silencing and overexpression to show that RrGT2 expression is correlated with anthocyanin accumulation in flowers in two species of rose, R. rugosa and R. davurica, and when RrGT2 is heterologously expressed in tobacco. They also characterized the expression of RrGT2 in different tissues and developmental stages in the two rose species using qRT-PCR experiments. Overall, I found that the manuscript was well-written and the experiments were sound with appropriate controls and complementary experiments. However, there are a few minor points that should be addressed before acceptance.

Lines 60 to 73: I am not very familiar with research in rose, but these sentences describe the current state of research in Rosa rugosa without citing any references. Citations need to be added or at least the authors should describe where all this information came from.

Fig 1B: The authors do not seem have stated what was aligned (cDNA or protein), and using which aligner software, to create the phylogenetic tree.

Fig 3: The statistical test used to test for significance is not included in the caption. Other qPCR figures appear to have this information.

Fig 5: Authors should consider adding a panel to indicate where in the anthocyanin pathway these ‘upstream’ genes come from.

Tables 1, 2, 3: Authors should consider representing these data using a bar chart. Table 1 in particular is quite dense with data and would be better visualized as a figure.

Line 470: It was surprising to me that the tissue expression pattern of RrGT2 in wildtype rose (Fig 3C,D) is so similar to the expression pattern when heterologously expressed in tobacco under the control of a constitutive 35S promoter (Fig 7E). Does this not suggest that RrGT2 is post-transcriptionally regulated to provide tissue specificity (at least in tobacco)? I don’t think there is any need, for the scope of this paper, to do any additional experiments to clarify this finding, but I thought this was an unexpected and interesting result that the authors did not elaborate on at all in the discussion.

Author Response

Dear reviewer #1

Thank you very much for your comments on our manuscript entitled “RrGT2, a key gene associated with anthocyanin biosynthesis in Rosa rugosa, was identified via virus-induced gene silencing and overexpression” (ijms-402595). We appreciate the comments and made sincere efforts to address them in the revised manuscript. The manuscript was carefully revised according to the comments and suggestions, and our point-by-point responses to the comments are listed below. Furthermore, The line number of ‘Response to Reviewer #1’ refers to that of the revised manuscript without track changes.

We sincerely hope that the revised manuscript meets the publication standards for International Journal of Molecular Sciences. Please do not hesitate to contact us should you have any further queries.

Reviewer # 1

Sui et al. present results from virus-induced gene silencing and overexpression to show that RrGT2 expression is correlated with anthocyanin accumulation in flowers in two species of rose, R. rugosa and R. davurica, and when RrGT2 is heterologously expressed in tobacco. They also characterized the expression of RrGT2 in different tissues and developmental stages in the two rose species using qRT-PCR experiments. Overall, I found that the manuscript was well-written and the experiments were sound with appropriate controls and complementary experiments. However, there are a few minor points that should be addressed before acceptance.
1. Lines 60 to 73: I am not very familiar with research in rose, but these sentences describe the current state of research in Rosa rugosa without citing any references. Citations need to be added or at least the authors should describe where all this information came from.

Response: Thank you very much for your suggestion. In accordance with your suggestion, we have supplemented the references in the corresponding section (Line 51-59).
2. Fig 1B: The authors do not seem have stated what was aligned (cDNA or protein), and using which aligner software, to create the phylogenetic tree.

Response: Thank you very much for your suggestion. And I'm sorry to have confused you because of our statement. The phylogenetic tree (Figure 1B) was constructed from the amino acid sequences of 21 plants, including the sequence of RrGT2, using MEGA 5.0 software (Line 117-119). At the same time, we have also revised the interpretation of the legend of Figure 1B (Line 126).

3. Fig 3: The statistical test used to test for significance is not included in the caption. Other qPCR figures appear to have this information.

Response: Thank you very much for your suggestion. We've added the caption of statistical analyses to the Materials and Methods chapter (Line 581-585).
4. Fig 5: Authors should consider adding a panel to indicate where in the anthocyanin pathway these ‘upstream’ genes come from.

Response: Thank you very much for your suggestion. According to your suggestion, we've drawn a panel about the metabolic pathway analysis of structural genes involved in anthocyanin biosynthesis named Figure 6.

5.Tables 1, 2, 3: Authors should consider representing these data using a bar chart. Table 1 in particular is quite dense with data and would be better visualized as a figure.

Response: Thank you very much for your suggestions. After fully considering your suggestions and combining our expectations and requirements for the presentation of relevant data, we decided to make the corresponding data in Table 1, 2, 3 as a bar chart (Figure 10) and presented it in the manuscript, and added the original data of Table 1, 2, 3 to the supplementary materials (Table S2, S3, S4).

6. Line 470: It was surprising to me that the tissue expression pattern of RrGT2 in wildtype rose (Fig 3C,D) is so similar to the expression pattern when heterologously expressed in tobacco under the control of a constitutive 35S promoter (Fig 7E). Does this not suggest that RrGT2 is post-transcriptionally regulated to provide tissue specificity (at least in tobacco)? I don’t think there is any need, for the scope of this paper, to do any additional experiments to clarify this finding, but I thought this was an unexpected and interesting result that the authors did not elaborate on at all in the discussion.
Response: First of all, thank you very much for your in-depth analysis and comments on this part. We have given serious thought to the question that you have raised and we have the following views on it:

Firstly, in theory, if 35S promoter is used for expression experiments, the expression of exogenous genes in the tissues of the over-expressed plants should be consistent or slightly different in general. However, in the actual results, we found that the expression level of RrGT2 gene varies greatly in different tissues. Therefore, we speculate that this may be due to the differences of expression of 35S promoter driven target gene in different tissues.

Secondly, in addition, we also speculate that the RrGT2 gene may have post-transcriptional regulation in different tissues of over-expressed tobacco plants, resulting in significant differences in the expression of RrGT2 gene in different tissues.

So, in our opinion, this was not a simple coincidence. Of course, according to previous studies, the expression of structural genes is often regulated, but as you said, our manuscript more focuses on the functional analysis of RrGT2 gene in anthocyanin biosynthesis. And the regulation mechanism related to RrGT2 gene will be the focus of our next research, and this issue has been briefly supplemented in the section under discussion (Line 443-445).

Reviewer 2 Report

Reduce the abstracts below 200 words

Reference section need to correct ( refer the pdf attached)

Author Response

Dear reviewer #2

Thank you very much for your comments on our manuscript entitled “RrGT2, a key gene associated with anthocyanin biosynthesis in Rosa rugosa, was identified via virus-induced gene silencing and overexpression” (ijms-402595). We appreciate the comments and made sincere efforts to address them in the revised manuscript. The manuscript was carefully revised according to the comments and suggestions, and our point-by-point responses to the comments are listed below. Furthermore, The line number of ‘Response to Reviewer #2’ refers to that of the revised manuscript without track changes.

We sincerely hope that the revised manuscript meets the publication standards for International Journal of Molecular Sciences. Please do not hesitate to contact us should you have any further queries.

Reviewer # 2

1. Reduce the abstracts below 200 words

Response: Thank you very much for your suggestion. We have revised the section of Abstract (Line 13-28) to within 200 words in accordance with your suggestion.

2. Reference section need to correct ( refer the pdf attached)

Response: Thank you very much for your suggestion again. We have added and revised the relevant references in the revised manuscript (Line 67-70 and more).

Reviewer 3 Report

In this study, the roles of RrGT2 in anthocyanins accumulation were described. It is clearly shown that RrGT2 plays a positive role. Its overexpression in tobacco also suggests the practical value of this gene. The reviewer especially likes the application of VIGS method in Rose plant (especially flower). Therefore, the reviewer suggests accepting this manuscript by IJMS, if the following suggested revisions could be accepted by the authors.

Major:

1.       The authors’ earlier publication in AJPS which reports the cloning of RrGT2 should be cited. Importantly, the earlier publication showed the sequence analysis and expression of RrGT2, which are shown here again. It is unbelievable that even some pictures are same here as the previous publication. I think the earlier publication should be cited and the sequence and expression analysis here should be deleted or briefly described.

2.       It is very interesting that the VIGS method was successfully applied in Rose even in field conditions. More details of the VIGS application should be shown, such as the environment temperature, the efficiency of this method in Rose.

3.       Overexpression of RrGT2 in tobacco is properly conducted and the results are reasonable. However, the anthocyanin biosynthetic genes of tobacco should be checked to make sure that RrGT2 is essential for the metabolites increase. Another confusion is that, when RrGT2 was silenced in overexpression tobacco, the anthocyanin content was reduced to a level which is even lower than wild type plant. From this point, the internal genes expression should be influenced. Therefore, it is important to test the tobacco anthocyanin biosynthetic genes.

4.       If possible, the in vitro enzyme assay could be conducted. However, I don’t insist in this data.

Minor:

1.       In line 59, “and some lack yellow…” should be “and some show yellow…”? Maybe I’m not correct because the unavailable of the reference.

2.       The name of the gene in R. davurica should be changed to RdGT2 but not still RrGT2.

3.       In Figure 7B, RrGT2 expression in T1 was used as 1 relatively. Is the statistical analysis performed to compared to T1? This is not correct because they should be compared with control. Same problem in Figure 7E.

Author Response

Dear reviewer #3

Thank you very much for your comments on our manuscript entitled “RrGT2, a key gene associated with anthocyanin biosynthesis in Rosa rugosa, was identified via virus-induced gene silencing and overexpression” (ijms-402595). We appreciate the comments and made sincere efforts to address them in the revised manuscript. The manuscript was carefully revised according to the comments and suggestions, and our point-by-point responses to the comments are listed below. Furthermore, The line number of ‘Response to Reviewer #3’ refers to that of the revised manuscript without track changes.

We sincerely hope that the revised manuscript meets the publication standards for International Journal of Molecular Sciences. Please do not hesitate to contact us should you have any further queries.

Reviewer #3: In this study, the roles of RrGT2 in anthocyanins accumulation were described. It is clearly shown that RrGT2 plays a positive role. Its overexpression in tobacco also suggests the practical value of this gene. The reviewer especially likes the application of VIGS method in Rose plant (especially flower). Therefore, the reviewer suggests accepting this manuscript by IJMS, if the following suggested revisions could be accepted by the authors.

Major:

1. The authors’ earlier publication in AJPS which reports the cloning of RrGT2 should be cited. Importantly, the earlier publication showed the sequence analysis and expression of RrGT2, which are shown here again. It is unbelievable that even some pictures are same here as the previous publication. I think the earlier publication should be cited and the sequence and expression analysis here should be deleted or briefly described.

Response: First of all, thank you very much for your questions and suggestions. When the RrGT2 gene was first discovered, we analyzed and predicted the biological function of the RrGT2 gene at the theoretical level, and published it in the AJPS journal. After designing experiments and deeply studying the function of RrGT2 gene, we thought it is very necessary to explain the function of RrGT2 in a more comprehensive and detailed way, so we wrote this manuscript. In order to avoid suspicion and to avoid unnecessary misunderstanding, we did not cite our previous article in this manuscript. But, obviously, we haven't handled this problem very well, and we apologize for the confusion. The previously published article has been cited in the revised manuscript, and the sequence and expression analysis of RrGT2 gene are briefly described. Compared with the earlier publication in AJPS, the varieties used in the analysis of RrGT2 expression patterns in this manuscript are different. In addition, although there is some duplication in sequence analysis, the proportion of the content is very small, if it is deleted, then the integrity of the overall content of the article may be affected, so we hope to get your understanding.

2. It is very interesting that the VIGS method was successfully applied in Rose even in field conditions. More details of the VIGS application should be shown, such as the environment temperature, the efficiency of this method in Rose.

Response: Thank you again for your valuable suggestions. As far as we know, there were no similar reports on the application of VIGS technique in Rosa plants under field conditions. Since there was no successful experience to learn from, we were not absolutely sure wether it would be success or not when we decided to use VIGS technique to try this experiment. However, we tried to overcome a series of problems in the course of experiment, such as the selection of inoculation period, the selection of inoculation method, the treatment after inoculation and so on. Considering the flowering period of R. rugosa and R. davurica and the requirement of optimal silence time by VIGS technique, we decided to inoculate R. rugosa and R. davurica on the 30th to 40th days (from mid-March to mid-April) before flowering. Therefore, in the field conditions, we could not guarantee that the environment temperature fully conforms to the use of VIGS standards, of course, this was our attempt after all. Of course, we also have measures to improve, for example, before setting a specific inoculation date, we will look into the weather for at least a week to avoid bad weather. In addition, we set the specific inoculation time between 2: 00 and 4: 00 in the afternoon, because during this time, the environment temperature is relatively high, which is more in line with the inoculation operation of VIGS. Another reason is that inoculation in the afternoon leads to a faster transition to night, making dark processing more real.Regarding the treatment after inoculation, we adopted the method of bagging and using black plastic bags, which can not only be treated in darkness, but also achieve the effect of heat preservation to a certain extent.

   With regard to the supplement of the method, we have modified it in the manuscript (Line 585-591), and on the issue of the efficiency of the infection, we have added the data to the supplementary material (Table S5).

3. Overexpression of RrGT2 in tobacco is properly conducted and the results are reasonable. However, the anthocyanin biosynthetic genes of tobacco should be checked to make sure that RrGT2 is essential for the metabolites increase. Another confusion is that, when RrGT2 was silenced in overexpression tobacco, the anthocyanin content was reduced to a level which is even lower than wild type plant. From this point, the internal genes expression should be influenced. Therefore, it is important to test the tobacco anthocyanin biosynthetic genes.

Response: Thank you very much for your valuable comments and suggestions. First of all, I am very sorry that we did not detect the structural genes associated with anthocyanin biosynthesis in tobacco, and our explanations are as follows:

   Firstly, RrGT2 gene is the downstream structural gene of anthocyanin biosynthesis pathway, and its glycosylation is considered to be the last step of anthocyanin biosynthesis with stable structure. According to previous studies, the expression of upstream structural genes is regulated by transcription factors. Therefore, in theory, RrGT2 gene, which is also a structural gene, has no effect on upstream structural genes during its overexpression.

   Secondly, in the overexpressed experiments, we set up a double-layer control experiment. One was blank control group, the other was empty vector group. It can be seen from the results that there was no significant difference in flower color between tobacco lines with empty vector group and blank control group, which indicated that the deepening of flower color of tobacco lines with overexpression of RrGT2 gene was caused by exogenous RrGT2 gene.

   Thirdly, when tobacco lines with RrGT2 gene were treated with VIGS, we also set up a double-layer control experiment, that is, blank control group and empty virus vector group. The results showed that the flower color of the empty virus vector group was basically the same as that of the blank control group, while the silenced tobacco lines showed a pale phenotype in flowers. Apparently, this was due to the specific silencing of the RrGT2 gene.

   Through double control, we can basically determine that the phenotypic changes in the treated plants were caused by overexpression or silencing of RrGT2 gene, although there may be changes in other structural genes. Nevertheless, it must be said that this was indeed one of the shortcomings of our experiment. So, we hope to have your understanding.

   However, it was also surprising to us that anthocyanin levels in the gene silencing group were lower than those in wild type plants. But it can be basically excluded that this has nothing to do with the expression of upstream genes, because VIGS can only silence genes with identical or partially identical sequences to the gene fragments embedded in the virus. Therefore, we speculate that this may be due to the silencing of GT genes in tobacco, which have high homology with RrGT2 gene and also associated with anthocyanin biosynthesis. But unfortunately, the number of genes in the GT family of tobacco is also very large, and there are few related studies, so we can not detect the GT genes in tobacco for the time being.

   The study of regulation mechanism related to RrGT2 gene will be our next research plan. In subsequent studies, we will continue to follow your advice and conduct more thorough and rigorous research in a responsible manner for scientific research. We hope to have your understanding again.

4. If possible, the in vitro enzyme assay could be conducted. However, I don’t insist in this data.

Response: Thank you very much for your suggestion. But I'm sorry, in this part of the experiment, we didn't test the enzyme activity in vitro. Because according to the current research on RrGT2 gene, we can not predict the substrates well and accurately, and we also need to construct and optimize the induction and detection system of enzyme activity in vitro. This requires a relatively long process of exploration. According to our experimental plan, the next step will be to study the transcriptional regulation mechanism of RrGT2 gene and the specificity of its substrate, so the detection of enzyme activity will be included. We hope to have your understanding.

Minor:

1.  In line 59, “and some lack yellow…” should be “and some show yellow…”? Maybe I’m not correct because the unavailable of the reference.

Response: Thank you very much for your question, and I’m sorry for the questions we have caused you because of our statement. But "lack" is really what we want to say. Because for R. rugosa, varieties whose flowers are yellow, bright red, orange and compound colors are relatively scarce. We hope to have your understanding.

2. The name of the gene in R. davurica should be changed to RdGT2 but not still RrGT2.

Response: Thank you very much for your suggestion. At your suggestion, we have revised the manuscript (Line 147-165). After we cloned the RrGT2 gene from R. rugosa ‘Zizhi’, we cloned a gene from the cDNA of R. davurica with the same primers and sequenced the gene sequence which was identical to that of RrGT2. This may be because R. davurica is the mother of R. rugosa ‘Zizhi’, so there is no difference in some genes. And in order not to cause confusion in the article, we uniformly named this gene ‘RrGT2’. But obviously, this was wrong. So thank you again for helping us correct this mistake.

3. In Figure 7B, RrGT2 expression in T1 was used as 1 relatively. Is the statistical analysis performed to compared to T1? This is not correct because they should be compared with control. Same problem in Figure 7E.

Response: Thank you very much for your questions and suggestions. Because RrGT2 gene was not detected in blank control group and empty vector group, the expression of RrGT2 was ‘0’. Therefore, in order to facilitate the comparison of the relative expression of RrGT2 gene in three transgenic tobacco lines, we decided to take the expression of RrGT2 gene in T1 line as ‘1’ in Figure 7B and 7E (Due to the increase in the number of figures, the original figure 7 has now become figure 8). Of course, the result we got was a relative amount of expression, not an absolute amount of expression. We apologize for the confusion caused to you and we hope to have your understanding.

Round 2

Reviewer 3 Report

The authors of this manuscript have made efficient corrections based on my suggestions. Overall, I think this revised manuscript is suitable to be published in IJMS.